# Dietary Probiotics Modulate Gut Barrier and Immune-Related Gene Expression and Histomorphology in Broiler Chickens under Non- and Pathogen-Challenged Conditions: A Meta-Analysis

**DOI:** 10.3390/ani13121970

**Published:** 2023-06-13

**Authors:** Fitra Yosi, Barbara U. Metzler-Zebeli

**Affiliations:** 1Unit Nutritional Physiology, Institute of Physiology, Pathophysiology, and Biophysics, Department of Biomedical Sciences, University of Veterinary Medicine Vienna, 1210 Vienna, Austria; fitra.yosi@vetmeduni.ac.at; 2Christian-Doppler Laboratory for Innovative Gut Health Concepts of Livestock, University of Veterinary Medicine Vienna, 1210 Vienna, Austria; 3Department of Animal Science, Faculty of Agriculture, University of Sriwijaya, Palembang 30662, Indonesia

**Keywords:** barrier function, broilers, gastrointestinal tract, growth, histomorphology, immune response, meta-analysis, pathogen, probiotics

## Abstract

**Simple Summary:**

The supplementation of diets for broiler chickens has increased due to the increasing demand of consumers for antibiotic-free broiler products. Nevertheless, the benefits of probiotics for intestinal barrier and immune functions, as well as on growth performance in chickens, are still controversially discussed. In performing a meta-analysis, we found that dietary supplementation with probiotics of various genera/species can enforce intestinal barrier function. Moreover, our meta-regressions indicated that in pathogen-challenged birds, probiotics might effectively help reduce gut inflammation by suppressing the expression of pro-inflammatory cytokines. Probiotics further sustained the intestinal histomorphology and hence digestive and absorptive processes in challenged and non-challenged chickens.

**Abstract:**

Data published in the literature about the favorable effects of dietary probiotics on gut health in broiler chickens are inconsistent. To obtain a more comprehensive understanding, we conducted a meta-analysis to assess the effects of probiotics on the gut barrier and immune-related gene expression, histomorphology, and growth in chickens that were either challenged or non-challenged with pathogens. From the 54 articles published between 2012 and 2022, subsets of data, separately for non-challenged and challenged conditions, for response variables were created. The mean dietary probiotic concentrations ranged from 4.7 to 6.2 and 4.7 to 7.2 log_10_ colony-forming unit/kg under non-challenged and challenged conditions, respectively. Probiotics increased the expression of genes for mucins and tight junction proteins in the jejunum and ileum at weeks 3 and 6. The stimulatory effect of probiotics on tight junction protein expression was partly stronger in challenged than in non-challenged birds. Meta-regressions also showed an anti-inflammatory effect of probiotics under challenged conditions by modulating the expression of cytokines. Probiotics improved villus height at certain ages in the small intestine while not influencing growth performance. Dietary metabolizable energy, crude protein, and days post-infection modified the effects of probiotics on the observed variables. Overall, meta-regressions support the beneficial effects of probiotics on gut integrity and structure in chickens.

## 1. Introduction

The use of antibiotic growth promoters in chicken farms has been banned in many countries worldwide. Probiotics are considered a promising alternative for livestock animals, including poultry, as they seem to exert a favorable effect on gut health [1]. To date, various microbial genera have been investigated for use as probiotics in poultry diets, including *Lactobacillus*, *Bacillus*, *Enterococcus*, *Bifidobacterium*, and *Saccharomyces* [1]. Several studies show potential beneficial effects of probiotics on growth performance, absorptive and secretory processes, as well as expression of genes related to host defense mechanisms, barrier function, and inflammation in broiler chickens [2,3,4]. For instance, dietary probiotics, such as *B. subtilis* and *B. pumilus*, have been shown to stimulate host defense mechanisms at the ileal epithelium by modulating tight junction protein expression in the grower and finisher phases [2]. Other probiotics, such as *L. acidophilus* and *L. plantarum*, have been reported to exert anti-inflammatory effects in the small intestine by moderating pro-inflammatory nuclear factor-kappa B (NF-kB) signaling, which, in turn, leads to lower transcript levels pro-inflammatory cytokines in the jejunum and ileum [3,4]. However, the reported effects of probiotics on the gut epithelial response in chickens are inconsistent [5,6]. Multiple factors may be behind the controversial findings, including direct (e.g., strain and level of probiotics) and indirect factors (e.g., age of birds, intestinal sampling spot, and health status). Although the relationship between dietary probiotics and gene expression levels related to intestinal integrity and immunity in chickens has been described in recent qualitative reviews [7,8], the variation in results of the dependent variable due to influencing factors cannot be assessed in this manner [9]. The conductance of a meta-analysis is considered the most suitable method to address this complexity by generalizing the overall treatment effect, in our case, the effect of probiotics, presented in published studies [10,11]. To obtain a more comprehensive understanding of the efficacy of probiotics, investigations on each response variable were performed separately between original studies with or without pathogen challenge. Therefore, the present meta-analysis aimed to investigate the effects of dietary supplementation of probiotics on the expression of genes associated with intestinal barrier function and immune response, histomorphology, and growth performance in broiler chickens under non-challenged or pathogen-challenged conditions. Furthermore, we assessed the effects of dietary metabolizable energy, crude protein, and days post-infection as additional predictors to obtain a more accurate prediction on the observed variables.

## 2. Materials and Methods

### 2.1. Literature Search

For the identification of original articles, a literature search was conducted using 5 public search generators, including Scopus, PubMed, Web of Science, Science Direct, and Google Scholar (Figure 1). Research articles investigating the effects of dietary probiotics on the expression levels of genes related to intestinal barrier function and immune response in broiler chickens that were published in scientific journals between January 2012 and July 2022 were considered for data extraction. To identify adequate articles, the following search terms were used in different combinations: probiotic, direct-fed microbes, gut, intestine, barrier function, gut permeability, gut integrity, tight junction proteins, immune response, inflammatory cytokines, gut inflammation, intestinal immunity, chicken, and broiler.

### 2.2. Selection of Studies

Stringent criteria were applied in the decision to exclude or include the research articles in the present meta-analysis (Figure 1). The quality assessment criteria used in this study included detailed information on probiotics (type and level of dietary probiotics), chicken strain, body weight and age of chickens, rearing period, and number of chickens per treatment, diet composition, experimental design, including randomization of treatments, description of statistical analysis, and intra-study error (if standard deviation was provided, then it was converted to standard error). Only probiotics that were administered via the diet were considered in this study. In addition, studies investigating the combined effects of dietary probiotics with other treatments on target parameters were also included. From these studies, only data for control and probiotic treatments were considered. Regarding gene expression measurements, only studies that applied quantitative real-time PCR analysis and in which the relative gene expression was calculated using the 2^−ddCt^ method were included. Moreover, only literature data from in vivo experiments was considered.

### 2.3. Construction of Database

After screening the literature, we identified 54 eligible research articles that met the quality criteria (Figure 1). A minimum requirement of 3 studies and 10 single observations (treatment means) along with the standard error (SE) for each dependent variable was set as requirement for calculating the combined effect size [10,11]. The main predictive variable was the dietary probiotic concentration. Information about the probiotic species used was mandatory. Reported expression levels of genes related to intestinal barrier function and immune response in various intestinal segments (e.g., duodenum, jejunum, ileum, and ceca) were extracted as dependent variables. Moreover, details provided on the chickens (strain, sex, age, and start body weight), experimental setup (experimental design, number of treatments, rearing period, number of chickens per group, and sampling days), pathogen challenge (species or strain of pathogen, administration route, and days post-infection), ingredients and nutritional composition of the diet, and gene expression analysis (e.g., reference genes) were extracted as probable additional prediction variables in the regression analysis. When available, histomorphology data such as villus height, crypt depth, and villus height/crypt depth ratio, as well as growth performance, including average daily feed intake (ADFI), average daily body weight gain (ADG), and feed conversion ratio (FCR) were also extracted. If data from the articles were presented in graphical form, they were extracted using Web Plot Digitizer software (Version 4.5; Ankit Rohatgi, Pacifica, CA, USA).

Two databases were constructed: one for data from research with pathogen challenge and the other for data from studies without pathogen challenge (Figure 1). The next step was to construct datasets for individual dependent variable categories separately for data with or without pathogen challenge, i.e., one dataset each for gut barrier and immune response-related gene expression, histomorphology measures (i.e., villus height, crypt depth, and villus height/crypt depth ratio), and growth performance (i.e., ADG, ADFI, and FCR). Datasets for gut barrier and immune-related gene expression and histomorphology were further subdivided; one sub-dataset was created for each gut segment. For each gut segment, the sub-sets were then grouped by age of the chicken. The dataset for growth performance was divided into sub-datasets based on the stage of the rearing period: starter (1–3 weeks), finisher (4–6 weeks), and overall (1–6 weeks) periods. As there were not enough studies available to investigate each probiotic strain or species separately, results for the various species/strains of probiotics were analyzed together in this meta-analysis. A reference list of the sub-datasets of broiler studies is presented in Appendix A.

The screening for the non-challenge studies showed that the minimum number of studies and observations for gene expression variables related to intestinal barrier function and immune response were fulfilled for the jejunum and ileum at weeks 3 and 6 of life. Adequate numbers of studies and observations for histomorphology variables were available for the jejunum and ileum at weeks 3 and 6 of life. For growth performance variables, the extracted data for the starter, finisher, and entire rearing period also met the requirement. For the studies with pathogen challenge, with respect to the expression of genes related to the intestinal barrier, the variables for the jejunum at weeks 2 to 5 of life, ileum at weeks 3 and 4 of life, and ceca at week 4 of life provided the required number of studies and observations. The variables related to the immune response met the requirement for the jejunum and ileum at weeks 2 to 4 and ceca at week 2 of life. For histomorphology variables, the minimum requirement of studies and observations existed for the data with pathogen challenge for the duodenum at week 5 of life, jejunum at weeks 2, 3, and 5 of life, and ileum at weeks 3 and 5 of life. Adequate numbers of studies and observations were also available for growth variables of starter, finisher, and overall periods. Only dependent variables that met the minimum requirements will be presented.

To create comparability among response variables across studies, the log_2_fold values for the gene expression data were calculated in each sub-dataset between control and probiotic treatment for un-challenged and pathogen-challenged data. Positive and negative log_2_fold values indicate increased and decreased expression, respectively. Data were processed and displayed as fold change, which was calculated using logarithmic scale to base 2. As dietary metabolizable energy (ME) and crude protein (CP) can affect nutritional metabolism and growth performance, these variables were included as additional predictor variables for both databases. Specifically for the data from pathogen challenge studies, days post-infection (DPI), defined as the interval from the first day of pathogen administration to sample collection, was also incorporated as an additional predictor.

### 2.4. Data Analysis

Descriptive statistics on the predictive variable (dietary probiotic concentration) and dependent variables (expression of gut barrier and immune-related genes in the jejunum, ileum, and ceca; histomorphology measures in the duodenum, jejunum, and ileum; and growth parameters) were performed separately for the dataset with or without pathogen challenge using the SAS MEANS procedure (version 9.4; SAS Inst. Inc., Cary, NC, USA), as previously described [10,11]. Mixed modeling of each dependent variable was established using the MIXED procedure similar to Metzler-Zebeli et al. [10,11].
$$Y_{ij}=\alpha_{0}+\beta_{1}X_{ij}+s_{i}+b_{i}X_{ij}+e_{ij}$$
where $Y_{ij}$ = expected outcome for the dependent variable Y observed at level j (j = 2, …, *n*) of the predictor variable X in the study i, whereas n is the number of treatment means in study i, $\alpha_{0}$ = overall intercept across all studies (fixed effect), $\beta_{1}$ = overall regression coefficient of Y on X across all studies (fixed effect), $X_{ij}$ = the value j of continuous variable X in study i, $s_{i}$ = random effect of the study i (i = 1, …), $b_{i}$ = the random effect of study i on the regression coefficient of Y on X in study i, and $e_{ij}$ = the unexplained error. Thus, the model’s random effect components consist of $s_{i}+b_{i}X_{ij}+e_{ij}$, and the distributions are displayed below as follows:$$e_{ij}\;\sim \;iid\;N\;\left(0,\;\sigma_{e}{}^{2}\right)\;\;\mathrm{and}\;\left[\frac{S_{i}}{b_{i}}\right]\;\sim \;iid\;N\;\left[\left(\frac{0}{0}\right),\;\Sigma \;\right],$$
which assumes that $e_{ij}$ is normally distributed with a mean of 0 and constant variance and that $s_{i}$ and $b_{i}$ are normally distributed, have means of 0, and Σ is their variance–covariance matrix:$$\Sigma =\left[\begin{array}{cc} \sigma_{s}{}^{2} & \sigma_{sb}\\ \sigma_{sb} & \sigma_{b}{}^{2} \end{array}\right]$$

As predictor variables for both study and dietary probiotic concentration were examined. The initial random effects included the slope and intercept based on the study and concentration of dietary probiotics. To prevent positive correlation between intercept and slope, an unstructured variance–covariance matrix (type = UN) was used [12]. The dependent variable was weighted by the inverse of its squared SE (SE of the treatment mean taken directly from the studies) to consider unequal variance between studies. The squared terms of the predictor variables were entered into the model to check for a quadratic relationship if significant (*p* < 0.05). The variance–covariance matrix, in this case, was modeled as variance components (TYPE = VC). For the current data set, there was no significant quadratic correlation; instead, the predictor and response variables showed only linear relationships. The GPLOT technique was used to display the data. To assess the quality of fit, estimates, root mean square error (RMSE), and R^2^ were calculated. For established relationships, alteration in the quantity of the dependent variables as affected by dietary probiotic concentration was shown for an assumed probiotic concentration in the diet of 4 log_10_ colony forming units (CFU)/kg.

We performed backward elimination analyses for the datasets with and without pathogen challenge to obtain more accurate predictions of the factors influencing the dependent variables that were affected by the dietary probiotic concentration [10,11]. This enabled us to simultaneously assess how the response variable was affected by the predictors of dietary probiotic concentration, dietary probiotic concentration squared, and dietary ME and CP level, as well as DPI specifically for pathogen challenge datasets. Consideration of variance inflation factors smaller than 10 (which presupposes no substantial multicollinearity among the tested predictor variables) for each continuous independent variable served to limit model over-parameterization [10,11].

## 3. Results

### 3.1. Database Description

The main characteristics of the 54 studies that met the selection criteria are presented in Appendix A. Of the 54 studies, 14 and 28 studies were without and with pathogen challenge, respectively, whereas 12 studies provided data for challenged and un-challenged conditions. Overall, nine different genera and various species within these genera were administrated as probiotics in the included studies (Figure 2): *Bacillus* (29 studies) and *Lactobacillus* (19 studies) were predominantly used, followed by *Enterococcus* (6 studies), *Saccharomyces* (4 studies), *Pediococcus* (4 studies), *Clostridium* (3 studies), *Bifidobacterium* (3 studies), *Paenibacillus* (2 studies), and *Streptococcus* (1 study). Eight different *Bacillus* species (*B. subtilis*, *B. licheniformis*, *B. coagulans*, *B. amyloliquefaciens*, *B. mesentericus*, *B. methylotrophicus*, *B. tequilensis*, and *B. pumilus*), nine species for *Lactobacillus* (*L. acidophilus*, *L. plantarum*, *L. fermentum*, *L. reuteri*, *L. casei*, *L. animalis*, *L. gallinarum*, *L. johnsonii*, and *L. salivarius*), three species for *Bifidobacterium* (*B. animalis*, *B. bifidum*, and *B. thermophilum*), two species each for *Enterococcus* and *Pediococcus* (*E. faecium*, *E. fecalis*, *P. acidilactici*, and *P. pentosaceus*), one species each for *Clostridium*, *Streptococcus*, *Paenibacillus*, and *Saccharomyces* (*C. butyricum*, *S. faecalis*, *P. polymyxa*, and *S. cerevisiae*) were administered. In addition, 36 studies used only one mono-species probiotic, 7 studies used more than one mono-species probiotic, 8 studies used multi-species probiotics, and 3 studies used both mono- and multi-species probiotics. The experimental diets were mainly composed of corn, wheat, barley, bran, rice, distiller grain, and sorghum, with soybean meal, fish meal, corn gluten meal, corn protein powder, rapeseed meal, peanut meal, and cottonseed meal as protein feedstuffs (Appendix A). The experimental diets did not contain other bioactive compounds. Dietary ME/CP ratios were constant, with a mean of 0.6 and 0.7 for the starter and finisher diets, respectively, for the various response variables.

Descriptive statistical results of the predictor and dependent variables for the database without pathogen challenge are presented in Appendix A. For these data, means of dietary probiotic concentrations across genera/species for the starter phase (1–3 weeks of age) ranged from 4.7 to 5.7 log_10_ CFU/kg, and those for the finisher phase (4–6 weeks of age) ranged from 5.7 to 6.2 log_10_ CFU/kg for the various categories of response variables. The means of dietary ME levels for the starter period were 12.3–12.5 MJ/kg, whereas those for the finisher period were 12.8–13.0 MJ/kg. Dietary CP levels for the starter and finisher phases showed means of 21.3–21.6% and 19.4–19.6%, respectively, for the various response variables.

The results of descriptive statistics for predictor variables and dependent variables of the database with pathogen challenge are presented in Appendix A. Several pathogens were included in the original studies, such as *Escherichia coli*, *Clostridium perfringens*, *Eimeria (E. tenella*, *E. maxima*, *E. acervulina*, *E. mivati*, *E. brunetti*, *E. mitis*, and *E. praecox*), *Salmonella* (*S. enteritidis*, *S. pullorum*, and *S. minnesota*), *Listeria monocytogenes*, as well as the fungi *Fusarium graminearum* and aflatoxins. For these data, the means of dietary probiotic concentrations across genera/species for the starter and finisher phases ranged from 4.6 to 5.6 log_10_ CFU/kg and 4.6 to 7.2 log_10_ CFU/kg, respectively, for the various categories of response variables. The respective means for the dietary ME level for starter and finisher periods were 12.0–12.6 MJ/kg and 12.4–12.8 MJ/kg for various dependent variables. The dietary CP levels in the starter phase showed a mean of 20.8–21.9%, whereas those in the finisher phase were 19.0–20.5% for a different category of response variables. In addition, the mean DPI for measuring gut barrier and immune gene expression for the starter and finisher ages ranged from 3.4 to 14.3 days and 7.6 to 28.7 days, respectively, for various intestinal segments. The mean DPI for the histomorphology variables were 5.2–10.3 days for the starter phase and 30.6–32.0 days for the finisher phase. For growth variables, the mean DPI for the starter and finisher ages were 10.8 and 33.2 days, respectively.

### 3.2. Probiotic Effects on Gut Barrier and Immune-Related Gene Expression

The results for the meta-regressions between probiotics and gut barrier and immune-related gene expression without pathogen challenge are presented in Table 1, whereas those with pathogen challenge can be found in Table 2. Irrespective of the pathogen challenge, most relationships between probiotics and gene expression levels were established for the jejunum and ileum.

Without the pathogen challenge (Table 1), increasing probiotic concentrations linearly increased the jejunal expression of *MUC2*, *ZO1*, *OCLN*, and *CLDN1* at week 3 of life (R^2^ = 0.32–0.46; *p* < 0.05). For a probiotic concentration of 4 log_10_ CFU/kg, this would correspond to an increase in expression levels of these genes by 0.21-, 0.08-, 0.34-, and 0.16-fold, respectively. Likewise, at 6 weeks of life, increasing probiotic concentrations linearly increased the jejunal expression of *MUC2* and *ZO1* (R^2^ = 0.42–0.45; *p* < 0.05). Accordingly, the administration of a probiotic concentration of 4 log_10_ CFU/kg in the diet would increase the jejunal *MUC2* and *ZO1* expression levels by 0.62- and 0.28-fold, respectively. In the ileum, increasing probiotic concentrations linearly increased the expression of *MUC2* and *OCLN* of life at week 3 of life and of *OCLN* at week 6 of life (R^2^ = 0.26–0.57; *p* < 0.05), which corresponds to an upregulation of the *MUC2* and *OCLN* expressions by 0.38-, 0.26- and 0.14-fold, respectively, for an assumed dietary probiotic concentration of 4 log_10_ CFU/kg.

Regarding the meta-regressions with data from the pathogen challenge (Table 2), a positive linear relationship could be established between jejunal *CLDN3* expression and probiotic concentration at week 2 of life (R^2^ = 0.97; *p* < 0.001). Here, an assumed dietary probiotic concentration of 4 log_10_ CFU/kg would increase the jejunal *CLDN3* expression by 0.41-fold. Meta-regressions showed that increasing probiotic concentrations linearly increased the jejunal *ZO1* expression from weeks 2 to 4 of life (R^2^ = 0.31–0.51; *p* < 0.05) and that of *OCLN* from week 3 of life (R^2^ = 0.28; *p =* 0.028). Likewise, dietary probiotics positively influenced the expression of *ZO1* and *OCLN* in the ileum at week 4 of life (R^2^ = 0.56–0.71; *p* < 0.05), amounting to an increase of 0.17- and 0.28-fold with an assumed probiotic concentration of 4 log_10_ CFU/kg, respectively. In the ceca, expression of *ZO1* linearly increased with increasing dietary probiotic concentrations at week 4 of life (R^2^ = 0.62; *p =* 0.007), which corresponded to a 0.48-fold increase with a probiotic concentration of 4 log_10_ CFU/kg. 

Under pathogen-challenged conditions, increasing dietary probiotic concentrations linearly decreased jejunal *IFNG* expression at week 2 of life (R^2^ = 0.82; *p* < 0.001; Table 2), which would correspond to a 0.15-fold decrease with a probiotic concentration of 4 log_10_ CFU/kg in the diet. Similarly, a negative linear relationship existed between the jejunal *IL1B* expression with increasing probiotic concentrations at weeks 2 and 3 of life (R^2^ = 0.53–0.63; *p* < 0.05). In contrast, a dietary probiotic concentration of 4 log_10_ CFU/kg would increase the jejunal *IL10* expression by 0.61-fold at week 3 of life (R^2^ = 0.54; *p =* 0.004). Moreover, expression of jejunal *IL6* and *TNFA* linearly decreased at week 3 of life (R^2^ = 0.35–0.45; *p* < 0.05), amounting to 0.12- and 0.10-fold, respectively, with a probiotic concentration of 4 log_10_ CFU/kg. Like in the jejunum, increasing concentrations of dietary probiotics linearly downregulated the expression of *TLR4* and *IFNG* in the ileum at weeks 2 and 3 of life, respectively (R^2^ = 0.71–0.75; *p =* 0.001; Table 2). At the cecal mucosa, higher probiotic concentrations decreased *IL6* expression (R^2^ = 0.31; *p =* 0.017; Table 2) but increased the expression of *IL10* (R^2^ = 0.47; *p =* 0.030) by 0.14- and 0.85-fold, respectively, at week 2 of life, with an assumed dietary probiotic concentration of 4 log_10_ CFU/kg.

### 3.3. Probiotic Effects on Gut Histomorphology

For the data without pathogen challenge (Table 3), increasing probiotic concentrations linearly increased jejunal villus height at weeks 3 and 6 of life (R^2^ = 0.28–0.66, *p* < 0.05), and the jejunal villus height/crypt depth ratio at week 3 of life (R^2^ = 0.42; *p =* 0.009). In the ileum, a similar positive linear relationship between the probiotic concentration and villus height was observed at week 6 of life (R^2^ = 0.58; *p* < 0.001) and ileal villus height/crypt depth ratio at week 3 and 6 of life (R^2^ = 0.41–0.65; *p* < 0.05). For the results with the pathogen challenge (Table 4), increasing probiotic concentrations linearly increased the villus height in the duodenum at week 5 of life (R^2^ = 0.53; *p =* 0.002). A similar relationship was found for the jejunal villus height at week 3 of life (R^2^ = 0.42; *p =* 0.005). Moreover, dietary probiotic concentrations showed a negative relationship with crypt depth (R^2^ = 0.28–0.71; *p* < 0.05) but a positive linear relationship with jejunal villus height/crypt depth ratio (R^2^ = 0.29–0.40; *p* < 0.05) at weeks 2, 3, and 5 of life. In the ileum, increasing probiotic concentrations linearly increased the crypt depth and decreased the villus height/crypt depth ratio at week 5 of life (R^2^ = 0.37–0.41; *p* < 0.05).

### 3.4. Probiotic Effects on Growth Performance

The meta-regression results for the growth performance in broiler chickens without and with pathogen challenges are presented in Table 5 and Table 6, respectively. Both under pathogen and non-pathogen challenges, dietary probiotics did not affect the ADFI, ADG, and FCR of broilers either in the starter, finisher, or overall phases.

### 3.5. Backward Elimination Analysis

The backward elimination analysis was conducted separately for data without (Table 7) and with pathogen challenge (Table 8 and Table 9). For the data of chickens without pathogen challenge, backward elimination analysis showed that dietary probiotic concentration was the main factor influencing the expression of *MUC2*, *ZO1*, and *OCLN* in jejunum and ileum and *CLDN1* in jejunum at week 3 of life ($R^{2}\;$= 0.36–0.57; *p* < 0.05). Moreover, increasing dietary ME levels counteracted the positive relationship between dietary probiotic concentration and jejunal *MUC2* expression at week 6 of life ($R^{2}\;$= 0.70; *p* < 0.05). In contrast, an increasing dietary CP level potentiated the increase in jejunal *OCLN* expression with increasing dietary probiotic concentrations at week 6 of life (R^2^ = 0.62; *p* < 0.05). The positive relationship between dietary probiotic concentration and jejunal *ZO1* expression was potentiated by dietary ME but counteracted by dietary CP at week 6 of life (R^2^ = 0.70; *p* < 0.05). Both dietary ME and CP levels counteracted the increase in ileal *ZO1* expression with increasing dietary probiotic concentrations at week 6 of life (R^2^ = 0.76; *p* < 0.05). For the gut histomorphology, backward elimination analysis showed that dietary probiotics were the only factor influencing the jejunal villus height at week 6 of life (R^2^ = 0.28; *p =* 0.02). A higher dietary ME level potentiated the increase in jejunal and ileal villus height/crypt depth ratio at week 3 of life (R^2^ = 0.68–0.72; *p* < 0.05) but counteracted the increase in ileal villus height/crypt depth ratio at week 6 of life (R^2^ = 0.76; *p* < 0.05) with higher concentrations of dietary probiotics. In addition, an increasing dietary CP level potentiated the positive relationship between dietary probiotic concentration and ileal villus height at week 6 of life (R^2^ = 0.72; *p* < 0.05).

Backward elimination analysis for data from studies with pathogen challenge showed that the dietary probiotics concentration was a major factor influencing the expressions of *CLDN3*, *IL6*, *IL10*, *IL1B*, *TNFA*, and *IFNG* either in the jejunum, ileum, or ceca at week 2 and 3 of life (R^2^ = 0.31–0.97; *p* < 0.05; Table 8). In addition, an increasing dietary ME level counteracted the positive relationship between dietary probiotic concentration and *ZO1* and *IL10* expression either in the jejunum, ileum, or ceca at weeks 2 and 4 of life (R^2^ = 0.79–0.87; *p* < 0.05). Further results showed that dietary CP level counteracted the increased expression of *ZO1*, *OCLN*, and *CLDN1* (R^2^ = 0.45–0.84; *p* < 0.05) as well as the decreased expression of *IL6*, *IL1B*, and *TLR4* (R^2^ = 0.68–0.95; *p* < 0.05) with higher dietary probiotic concentrations in both the jejunum and ileum from weeks 2 to 4 of life. The positive relationship between dietary probiotic concentration and jejunal *ZO1* expression was counteracted by a higher dietary ME but potentiated by increasing dietary CP at week 2 of life (R^2^ = 0.81; *p* < 0.05). Both dietary ME and CP levels counteracted the increase in jejunal *ZO1* expression with higher concentrations of dietary probiotics at week 3 of life (R^2^ = 0.76; *p* < 0.05). Increasing DPI potentiated increased cecal *ZO1* and *IL10* expression (R^2^ = 0.79–0.96; *p* < 0.05) and decreased jejunal *IL6* expression (R^2^ = 0.77; *p* < 0.05) with increasing dietary probiotic concentrations at weeks 2 and 4 of life.

For the gut histomorphology (Table 9), backward elimination analysis indicated that dietary probiotic concentration was the only factor influencing the villus height/crypt depth ratio in the jejunum and ileum at weeks 2 and 5 of life (R^2^ = 0.38–0.41; *p* < 0.05). A higher dietary ME level potentiated an increase in the jejunal villus height (R^2^ = 0.77; *p* < 0.05) but counteracted the decrease in jejunal crypt depth (R^2^ = 0.62; *p* < 0.05), with higher concentrations of dietary probiotics at week 3 or 5 of life. In contrast, increasing dietary CP levels counteracted the increase in jejunal villus height/crypt depth ratio (R^2^ = 0.60; *p* < 0.05) but potentiated the decrease in jejunal crypt depth (R^2^ = 0.51–88; *p* < 0.05) with increasing dietary probiotic concentrations at weeks 2 and 3 of life. Increasing DPI potentiated the increase in villus height in the duodenum and ileum at weeks 3 and 5 of life (R^2^ = 0.75–0.77; *p* < 0.05) but counteracted the increase in the jejunal villus height/crypt depth ratio and ileal crypt depth at week 5 of life (R^2^ = 0.57–0.75; *p* < 0.05) with increasing dietary probiotic concentrations. 

## 4. Discussion

Factors such as type and dosage, chicken breed, rearing stage, the composition of the basal diet, and the health status of the bird can influence the physiological effects of probiotics in chickens, adding to the variation among individual studies. Due to that, literature results on the ability of dietary probiotics to modulate the expression of genes related to immune response and barrier function in the gastrointestinal tract of broiler chickens are inconsistent [5,13,14]. Likewise, the effects of dietary probiotics on changes in histo-morphological parameters of the small intestine and performance in chickens also vary [15,16,17,18]. The original research included in this meta-analysis covers a wide scope of experimental settings, which should enable inferring predictions for the effect of probiotics on the target variables. However, it needs to be noted that the present meta-regressions only provide general trends for probiotic use in chicken diets. The data available for the individual probiotics did not meet the minimum requirements. Therefore, the data for the single and multi-species probiotics from the individual studies were combined to perform the meta-regression analysis. A similar limitation existed for the pathogens and aflatoxins administrated in the challenge studies. It also needs to be kept in mind that there is a chance that studies with no or adverse effects of probiotics were not published. From the parameters that met the minimum selection criteria, meta-regressions support the effectiveness of probiotics in sustaining small intestinal and cecal barrier function as well as structural components under non-challenged and challenged conditions while also controlling pro-inflammatory signaling under challenged conditions. The meta-regressions also supported that probiotics may effectively counteract potential damage caused by pathogens or mycotoxins in the lower part of the small intestine, such as oxidative stress and compromised barrier function. Regressions further indicated a beneficial effect of probiotics on absorptive and secretory functions by increasing villus height and decreasing crypt depth in the small intestine, especially under pathogen-challenged conditions. Our results also provided evidence for the gut segment- and age-specific effects. However, it needs to be kept in mind that sufficient data were not always available for the same parameters at the various ages of the birds. Consequently, our results provide a general idea about target variables that were modified by the addition of probiotics in the grower-finisher phase.

Mechanistically, there are several potential modes of action on how the probiotics can influence mucosal gene expression, depending on the actual species and strain of probiotics used. The administrated probiotics across the included non-challenge and challenge studies were *Bacillus*, *Lactobacillus*, *Clostridium*, *Pediococcus*, *Bifidobacterium*, *Streptococcus*, *Paenibacillus*, *Enterococcus*, and *Saccharomyces*. Bacteria interact with the host via microbial metabolites and microbe-associated molecular patterns, which represent specific cell surface structures [19,20]. Consequently, we can assume that parts of the mucosal signaling may have been mediated via the activation of G protein-coupled receptors, pattern recognition receptors, and microbe–microbe interactions, including the production of antimicrobial and fermentation metabolites [21,22,23]. Across the various species, the present meta-regressions supported the anti-inflammatory effects of probiotics under challenged conditions, which may have subsequently contributed to the upregulation of the mucosal barrier, including the expression of tight junction proteins and other first line of defense genes. Certain G protein-coupled receptors sense fermentation end products, such as short-chain and medium-chain fatty acids [24,25]. Due to the lack of data from the original studies, we can only speculate about the fermentation acids that changed locally in the gut due to the probiotic supplementation. *Lactobacillus*, *Enterococcus*, *Pediococcus*, *Streptococcus*, *Paenibacillus*, *Bifidobacterium*, and *Bacillus* produce lactic acid as a major fermentation product, but depending on the strain, they also produce short-chain fatty acids [26,27]. *Clostridium* is probably mainly signaled via short-chain fatty acids [28,29]. Short-chain fatty acid-induced G protein-coupled receptor activation may decrease the expression of pro-inflammatory cytokines via the inhibition of *NFKB* expression [21]. Unfortunately, we could not extract sufficient data to assess the probiotic effect on *NFKB* expression under un-challenged and challenged conditions as well as on cytokine expression in non-challenged chickens. Nevertheless, moderation of the activation of the pro-inflammatory NF-kB signaling pathway may be behind the present findings for negative effects of probiotics on the expression of *IL1B* and *INFG* at the jejunal mucosa in week 2 of age and expression of *IL1B*, *IL6*, and *TNFA* in the challenged birds at week 3 of age. Moreover, based on the coefficient of determination for the cytokine expression under challenged conditions, probiotics seemed to be very efficient in the jejunum at week 2 of age and in the ileum at week 3 of age in the challenged chickens. Simultaneously, probiotics may act as anti-inflammatory agent by upregulating the expression of *IL10* in innate and adaptive immune cells [30], as indicated by the present results for the jejunum at week 3 and ceca at week 2 of age. Moreover, *Bacillus*-based probiotics may not only act as an anti-inflammatory agent via fermentation acids but by producing quorum-sensing peptides, such as competence and sporulation factor, which signals via the Akt and p38 MAPK pathways [31,32]. *Saccharomyces*-based probiotics, especially *Saccharomyces cereviceae*, have been shown to effectively suppress inflammation by binding certain pathogens and toxins via mannose residues on their cell surface. This may be behind the efficacy of *Sacharomyces* to control *Escherichia coli* and *Salmonella* spp. as well as and mitigate the effects of *Fusarium*-produced mycotoxins [33,34,35], which were the harmful agent used in the respective challenge studies.

Another mode of action in how fermentation metabolites (especially butyrate) can modulate pro-inflammatory signaling pathways is via inhibition of histone deacetylases in macrophages and dendritic cells [22,36]. From the included probiotics, mainly *Clostridium butyricum* produces butyrate [37,38,39]. The other genera as lactic acid-producing bacteria may have increased the intestinal butyrate levels via cross-feeding [40,41] and hence indirectly affected the activity of histone deacetylases and modified the expression of pro- and anti-inflammatory cytokines as well as of genes related to the barrier function and host secretions. In the absence of actual data for intestinal butyrate levels, however, we can only speculate whether the presence of the probiotics led to physiologically relevant changes in intestinal butyrate production. Aside from interacting directly with the host, it can be assumed that part of the observed effects was mediated via the interaction of the probiotics with the commensal microbiota through fermentation acids and antimicrobials [42,43]. The latter metabolites can help shape the overall microbiota composition and inhibit the proliferation of pathogens and/or the expression of virulence factors [44,45]. For instance, reuterin produced by *Limosilactobacillus reuteri* is effective to control dysbiosis [46,47]. Similarly, antimicrobial compounds produced by certain *Bacillus*-based probiotics, such as surfactin, iturin, and fengycin, have also been reported to be effective against pathogenic bacteria [48]. Any alteration in the microbial composition automatically changes the composition of the microbial cell surface structures, which are recognized by pattern-recognition receptors at the gut mucosa and immune cells [49]. Unfortunately, not much data were available for the expression of pattern-recognition receptors in the included studies. In pathogen-challenged birds, our meta-regressions support a downregulating effect of the probiotics on *TLR4* expression in the ileum at week 2 of age. In the respective original studies, the pathogens that were administrated were Gram-negative bacteria, such as *Escherichia coli* and *Salmonella* spp. [50,51], which comprise highly immune-reactive lipopolysaccharide recognized by TLR-4 [20]. This finding may indicate that probiotics effectively inhibited the proliferation of the administrated pathogens and/or moderated the TLR-4 activation. Harmful agents, such as *Eimeria*, fungi, and mycotoxins, likely signaled via different pattern recognition receptors than TLR-4. In general, it is thinkable that probiotics mediated their anti-inflammatory effect via lower ligand-specific activation of the respective pattern recognition receptors. This, in turn, probably led to a lower *NFkB* expression and/or gene expression within the AMP-activated protein kinase, MAPK, or Akt-signaling pathways [52,53], and ultimately to a downregulation in expression levels of pro-inflammatory cytokines (e.g., *IL1B*, *IL6*, *INFG*, and *TNFA*) at the investigated gut sites.

The literature results suggested a protective effect of probiotics on intestinal integrity due to increased mucus production [54,55] and by stimulating the expression of tight junction proteins [55,56,57]. The present meta-regressions confirmed this assumption. However, fewer data were available for *MUC2* expression from the challenge studies; therefore, the present findings mainly support the beneficial effects of probiotics in non-challenged birds. Moreover, the stimulating effect of probiotics on the *MUC2* expression seemed to last longer in the jejunum than in the ileum of chickens under non-challenged conditions, which may be related to the length of the small intestine and age-related maturation of the immune system in the older chicken [42,58]. The aforementioned effects of probiotics on lower pro-inflammatory cytokine expression may explain their stimulatory effect on the expressions of *CLDN3*, *OCLN*, and *ZO1* in the jejunum, *OCLN* and *ZO1* in the ileum, and *ZO1* in the ceca at week 2, 3, or 4 of age. However, the stimulatory effect was not consistent for all available tight junction protein genes, especially for the claudin genes, which might be related to developmental changes in the gut epithelial functioning and the actual role of the tight junction protein, which needs further investigation. When comparing the non-challenged with the challenged conditions, our meta-regressions indicated an upregulation of the expression of *CLDN1* by the probiotics in non-challenged birds at week 3 of age. Under challenged conditions, however, probiotics did not modify the transcription of *CLDN1* but that of *CLDN3* at week 2 of age.

In individual studies, probiotics were shown to modulate gut histo-morphological parameters [37,59,60]. Our meta-regressions confirm that probiotics can effectively increase villus height and villus height/crypt ratio in non-pathogen- and pathogen-challenged conditions. Probiotics may increase villus height by inducing mitotic cell division and promoting epithelial cell proliferation [61]. Longer villi are associated with improved digestive and absorptive capabilities at the small intestinal mucosa [61]. In addition, probiotics seemed to have a stronger effect on crypt depth under pathogen-challenged conditions in both jejunum and ileum. A shallower crypt is associated with slower cell turnover [62], potentially indicating that the probiotics prevented the disruption of epithelial cells due to the administrated pathogens.

The backward elimination analysis was helpful in the assessment of the impact of certain dietary effects on the target variables. According to the best-fit model, higher dietary ME and CP levels were important influential factors that counteracted the efficacy of probiotics to increase the expression of *MUC2*, tight junction proteins, and anti-inflammatory cytokines and decrease pro-inflammatory cytokines in the small intestine and ceca. For instance, higher dietary CP may act pro-inflammatory in birds under challenged conditions by stimulating the proliferation of proteolytic taxa in the gut, which could lead to the activation of *TLR4* expression. Of the administrated pathogens and toxins, *Salmonella* and *Escherichia coli* as Gram-negative and proteolytic bacteria, for instance, may have benefited from increased dietary CP levels. Higher dietary ME, most often caused by a higher starch content of the diet, has been shown to reduce the number of butyric acid-producing bacteria and increase Gram-negative bacteria [63,64], which may act as a pro-inflammatory agent. In contrast to the finding at the gene expression level, the best-fit model also indicated that higher levels of dietary ME and CP could enhance the effect of probiotics on intestinal villus height, which may be related to the stimulation of growth and proliferation of intestinal epithelial cells due to greater nutrient availability [65]. The backward elimination analysis also suggested a certain recovery of the gut mucosa after the pathogen challenge that was independent of the probiotics. Accordingly, with increasing time post-infection, the expression levels of genes for pro-inflammatory cytokines decreased, whereas those of genes coding for anti-inflammatory cytokines and tight junction proteins increased.

## 5. Conclusions

This present meta-analysis confirmed the results from individual studies at the gene expression level that probiotics can support intestinal barrier function in the small intestine under non-pathogen-challenged conditions in broiler chickens. From the available data that were used in this present analysis, it can be further deduced that under challenged conditions with pathogens and mycotoxins, probiotics do not only increase the expression of barrier function genes, but they mediate anti-inflammatory effects via modulation of cytokine expression in the small intestine and ceca. The effect of probiotics was not limited to the changes in gene expression but was also detectable at the structural level, where they improved villus height and crypt depth and hence influenced absorptive and secretory processes at the small intestinal epithelium. However, the present meta-regressions did not support the effect of probiotics on growth performance. Other sources of variation that could potentially influence or counteract the effects of probiotics in the diet included the dietary levels of ME and CP as well as the DPI in the challenge studies. Limitations of this present meta-analysis were that insufficient data were available from individual studies for the various probiotics and administrated pathogens and mycotoxins. Therefore, the present meta-regressions provide general trends that should be verified in the future when more data for the various single and multi-strain probiotics are available.

## Figures and Tables

**Figure 1 animals-13-01970-f001:**
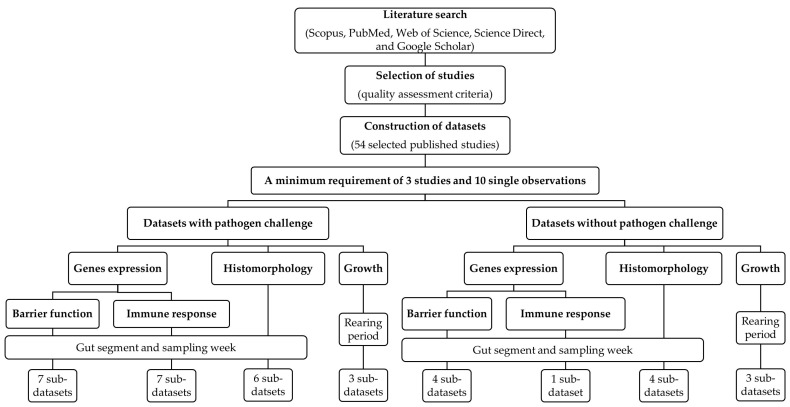
Flowchart showing the process of the collection and selection of original studies as well as the construction of databases in the current meta-analysis.

**Figure 2 animals-13-01970-f002:**
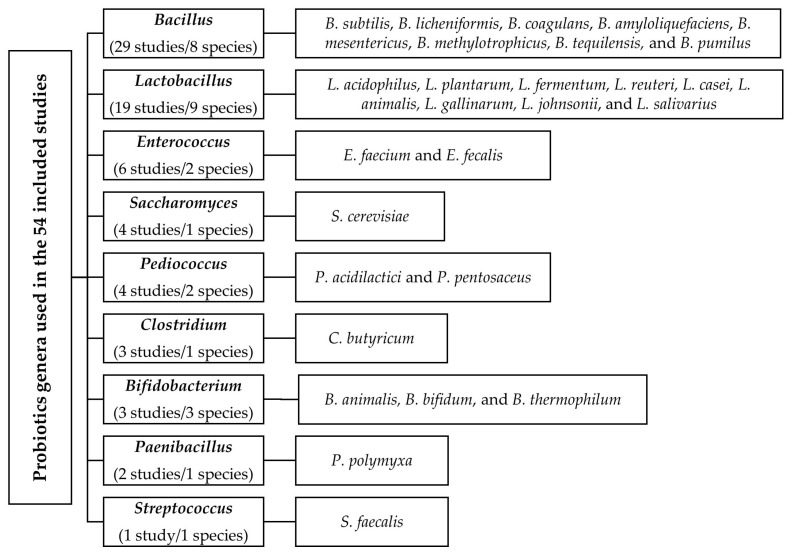
Overview of genera and species within genera administered as probiotics in the included studies.

**Table 1 animals-13-01970-t001:** Prediction of relative expression (fold change) of jejunal and ileal expression of genes related to gut barrier function and immune response in broiler chickens at weeks 3 to 6 of life without pathogen challenge.

Response Variable (Y) ^1,2^	n_Treat_	Parameter Estimates				Model Statistics		
		Intercept	SE_Intercept_	Slope	SE_Slope_	RMSE	R^2^	*p*-Value
Jejunum, Week 3								
*MUC2*	10	0.95	0.149	0.053	0.023	0.336	0.40	0.049
*ZO1*	11	0.99	0.046	0.019	0.007	0.104	0.46	0.023
*OCLN*	13	0.96	0.247	0.084	0.037	0.612	0.32	0.044
*CLDN1*	11	0.99	0.111	0.041	0.017	0.251	0.39	0.041
*IL1B*	11	1.01	0.060	−0.012	0.009	0.135	0.15	0.241
*IFNG*	14	1.02	0.058	−0.003	0.009	0.154	0.01	0.773
*TLR2*	11	1.07	0.181	0.014	0.028	0.407	0.03	0.621
Jejunum, Week 6								
*MUC2*	10	0.99	0.449	0.156	0.065	0.902	0.42	0.044
*ZO1*	14	1.00	0.170	0.069	0.022	0.382	0.45	0.009
*OCLN*	16	0.94	0.245	0.067	0.032	0.604	0.24	0.056
*CLDN1*	12	1.00	0.266	0.060	0.034	0.533	0.23	0.112
Ileum, Week 3								
*MUC2*	10	0.89	0.217	0.095	0.030	0.439	0.57	0.012
*ZO1*	11	0.97	0.171	−0.016	0.023	0.347	0.05	0.510
*OCLN*	13	0.85	0.150	0.064	0.020	0.338	0.47	0.009
*CLDN1*	10	0.96	0.115	0.036	0.016	0.233	0.39	0.054
Ileum, Week 6								
*MUC2*	15	0.98	0.538	0.130	0.074	1.211	0.19	0.103
*ZO1*	14	0.86	0.463	0.127	0.062	0.930	0.26	0.061
*OCLN*	16	0.97	0.115	0.034	0.015	0.259	0.26	0.043
*CLDN1*	11	0.94	0.362	0.087	0.049	0.728	0.26	0.112

n_Treat_, number of treatment means; SE, standard error; RMSE, root mean square error; *MUC2*, mucin−2; *ZO1*, zonula occludens-1; *OCLN*, occludin; *CLDN1*, claudin-1; *IL1B*, interleukin-1beta; *TLR2*, Toll-like receptor-2; *IFNG*, interferon-gamma. ^1.^ Probiotic genera included for these response variables were *Bacillus*, *Bifidobacterium*, *Lactobacillus*, *Clostridium*, *Enterococcus*, *Pediococcus*, *Paenibacillus*, and *Saccharomyces*. ^2.^ Data were calculated as log_2_fold change between probiotic and control treatments and expressed in fold change using a logarithmic scale to base 2.

**Table 2 animals-13-01970-t002:** Prediction of relative expression (fold change) of jejunal, ileal, and cecal expression of genes related to barrier function and immune response in broiler chickens from weeks 2 to 5 of life with pathogen challenge.

Response Variable (Y) ^1,2,3,4^	n_Treat_	Parameter Estimates				Model Statistics		
		Intercept	SE_Intercept_	Slope	SE_Slope_	RMSE	R^2^	*p*-Value
Jejunum, Week 2								
*ZO1*	14	0.99	0.026	0.015	0.004	0.070	0.51	0.004
*OCLN*	16	0.71	0.909	0.170	0.137	2.420	0.10	0.234
*CLDN1*	14	0.94	0.226	0.041	0.035	0.601	0.10	0.264
*CLDN3*	10	1.00	0.040	0.103	0.007	0.089	0.97	<0.001
*IL1B*	10	1.00	0.015	−0.009	0.003	0.035	0.63	0.006
*IL10*	14	1.04	0.247	0.015	0.039	0.606	0.01	0.707
*IFNG*	10	1.00	0.037	−0.037	0.006	0.083	0.82	<0.001
Jejunum, Week 3								
*MUC2*	10	0.93	0.238	0.051	0.037	0.538	0.19	0.205
*ZO1*	17	0.97	0.098	0.036	0.014	0.260	0.31	0.021
*OCLN*	17	0.94	0.177	0.062	0.025	0.473	0.28	0.028
*CLDN1*	14	0.80	0.617	0.138	0.091	1.524	0.16	0.155
*IL1B*	17	1.00	0.074	−0.042	0.010	0.198	0.53	0.001
*IL6*	12	1.01	0.096	−0.030	0.013	0.216	0.35	0.044
*IL10*	13	0.91	0.290	0.152	0.042	0.719	0.54	0.004
*IFNG*	18	1.00	0.106	−0.022	0.016	0.303	0.11	0.190
*TNFA*	10	1.01	0.066	−0.026	0.010	0.150	0.45	0.033
Jejunum, Week 4								
*ZO1*	12	1.06	0.085	0.034	0.013	0.192	0.40	0.026
*OCLN*	12	0.99	0.283	0.056	0.043	0.634	0.15	0.220
*IL1B*	10	0.98	0.113	0.076	0.017	0.227	0.72	0.002
*IFNG*	14	0.99	0.186	0.038	0.028	0.458	0.13	0.198
Jejunum, Week 5								
*MUC2*	13	0.97	0.206	−0.004	0.026	0.358	0	0.890
Ileum, Week 2								
*IFNG*	10	1.04	0.231	−0.015	0.036	0.517	0.02	0.677
*TLR4*	10	1.03	0.048	−0.035	0.007	0.107	0.75	0.001
Ileum, Week 3								
*ZO1*	16	0.96	0.196	0.050	0.028	0.483	0.19	0.096
*OCLN*	16	0.98	0.207	0.056	0.029	0.509	0.21	0.077
*CLDN1*	11	0.98	0.239	0.010	0.035	0.479	0.01	0.785
*IL10*	10	0.97	0.178	0.013	0.026	0.358	0.03	0.626
*IFNG*	12	1.01	0.043	−0.032	0.006	0.097	0.71	0.001
Ileum, Week 4								
*ZO1*	11	0.98	0.062	0.042	0.009	0.141	0.71	0.001
*OCLN*	11	0.93	0.143	0.070	0.021	0.325	0.56	0.008
*CLDN1*	11	0.89	0.263	0.082	0.038	0.597	0.34	0.059
*TNFA*	10	0.95	0.121	−0.023	0.016	0.244	0.21	0.185
Ceca, Week 2								
*IL6*	18	0.96	0.093	−0.034	0.013	0.270	0.31	0.017
*IL8*	10	0.83	0.251	−0.001	0.031	0.518	0	0.967
*IL10*	10	1.12	0.571	0.213	0.081	1.014	0.47	0.030
Ceca, Week 4								
*ZO1*	10	1.21	0.198	0.119	0.033	0.401	0.62	0.007

n_Treat_, mean number of treatments; SE, standard error; RMSE, root mean square error; *MUC2*, mucin-2; *ZO1*, zonula occludens-1; *OCLN*, occludin; *CLDN1,-3*, claudin-1,-3; *IL6,-8,-10,-1B*, interleukin-6,-8,-10,-1beta; *TLR4*, Toll-like receptor-4; *IFNG*, interferon-gamma; *TNFA*, tumor necrosis factor-alpha. ^1.^ Probiotic genera included for these response variables were *Bacillus*, *Bifidobacterium*, *Lactobacillus*, *Paenibacillus*, *Clostridium*, *Enterococcus*, *Pediococcus*, *Streptococcus*, and *Saccharomyces*. ^2.^ Pathogens included for these response variables were *E. coli*, *C. perfringens*, *S. enteritidis*, *E. maxima*, *E. tenella*, *E. acervulina*, *E. mivati*, *E. brunetti*, *E. mitis*, *E. praecox*, *F. graminearum*, *S. pullorum*, *S. minnesota*, *L. monocytogenes*, and Aflatoxin B1. ^3.^ Means of days post-infection ranged from 3.4 to 28.7 days for various ages and gut segments. ^4.^ Data were calculated as log_2_fold change between probiotic and control treatments and expressed in fold change using a logarithmic scale to base 2.

**Table 3 animals-13-01970-t003:** Prediction of jejunal and ileal histomorphology (fold change) in broiler chickens at weeks 3 and 6 of life without pathogen challenge.

Response Variable (Y) ^1,2^	n_Treat_	Parameter Estimates				Model Statistics		
		Intercept	SE_Intercept_	Slope	SE_Slope_	RMSE	R^2^	*p*-Value
Jejunum, Week 3								
Villus Height	15	1.00	0.030	0.022	0.004	0.080	0.66	<0.001
Crypt Depth	15	0.99	0.041	−0.005	0.006	0.110	0.05	0.411
Villus Height/Crypt Depth	15	1.00	0.064	0.029	0.009	0.171	0.42	0.009
Jejunum, Week 6								
Villus Height	19	1.00	0.032	0.011	0.004	0.084	0.28	0.020
Crypt Depth	19	1.00	0.061	0.008	0.008	0.163	0.05	0.348
Villus Height/Crypt Depth	19	1.01	0.050	0.004	0.007	0.133	0.02	0.529
Ileum, Week 3								
Villus Height	11	0.98	0.037	0.003	0.005	0.083	0.03	0.585
Crypt Depth	11	1.00	0.058	−0.016	0.008	0.130	0.29	0.088
Villus Height/Crypt Depth	11	0.99	0.065	0.023	0.009	0.147	0.41	0.034
Ileum, Week 6								
Villus Height	17	0.99	0.023	0.014	0.003	0.058	0.58	0.000
Crypt Depth	17	1.00	0.048	−0.004	0.006	0.119	0.02	0.570
Villus Height/Crypt Depth	17	1.01	0.022	0.015	0.003	0.055	0.65	<0.001

n_Treat_, number of treatment means; SE, standard error; RMSE, root mean square error. ^1.^ Probiotic genera included for these response variables were *Bacillus*, *Bifidobacterium*, *Lactobacillus*, *Clostridium*, *Enterococcus*, and *Saccharomyces.* ^2.^ Data were calculated as log_2_fold change between probiotic and control treatments and expressed in fold-change using a logarithmic scale to base 2.

**Table 4 animals-13-01970-t004:** Prediction of duodenal, jejunal, and ileal histomorphology (fold change) in broiler chickens from weeks 2 to 5 of life with pathogen challenge.

Response Variable (Y) ^1,2,3,4^	n_Treat_	Parameter Estimates				Model Statistics		
		Intercept	SE_Intercept_	Slope	SE_Slope_	RMSE	R^2^	*p*-Value
Duodenum, Week 5								
Villus Height	15	1.01	0.026	0.013	0.003	0.046	0.53	0.002
Crypt Depth	15	0.99	0.037	0.008	0.005	0.063	0.18	0.117
Villus Height/Crypt Depth	15	1.01	0.042	0.002	0.005	0.073	0.01	0.776
Jejunum, Week 2								
Villus Height	11	0.98	0.050	0.012	0.007	0.113	0.23	0.132
Crypt Depth	11	1.00	0.020	−0.014	0.003	0.046	0.71	0.001
Villus Height/Crypt Depth	11	0.98	0.088	0.029	0.013	0.197	0.38	0.044
Jejunum, Week 3								
Villus Height	17	0.99	0.040	0.019	0.006	0.106	0.42	0.005
Crypt Depth	17	1.01	0.035	−0.012	0.005	0.093	0.29	0.027
Villus Height/Crypt Depth	17	0.97	0.085	0.038	0.012	0.226	0.40	0.007
Jejunum, Week 5								
Villus Height	17	1.00	0.026	0.005	0.003	0.052	0.13	0.148
Crypt Depth	17	0.99	0.024	−0.007	0.003	0.049	0.28	0.029
Villus Height/Crypt Depth	17	1.01	0.030	0.009	0.004	0.059	0.29	0.025
Ileum, Week 3								
Villus Height	13	1.01	0.020	0.006	0.003	0.046	0.30	0.052
Crypt Depth	13	1.00	0.057	0.007	0.008	0.127	0.06	0.434
Villus Height/Crypt Depth	13	1.00	0.035	−0.001	0.005	0.077	0	0.873
Ileum, Week 5								
Villus Height	15	0.99	0.055	0.000	0.007	0.095	0	0.962
Crypt Depth	15	0.99	0.030	0.010	0.004	0.052	0.37	0.016
Villus Height/Crypt Depth	15	1.00	0.025	−0.009	0.003	0.043	0.41	0.011

n_Treat_, number of treatments means; SE, standard error; RMSE, root mean square error. ^1^ Probiotic genera included for these response variables were *Bacillus*, *Lactobacillus*, *Enterococcus*, and *Saccharomyces.*
^2.^ Pathogens included for these response variables were *E. coli*, *C. perfringens*, *S. enteritidis*, *E. maxima*, *E. tenella*, *E. acervulina*, *E. mitis*, *E. praecox*, and *F. graminearum.*
^3.^ Means of days post-infection ranged from 5.2 to 32.0 days for various ages and gut segments. ^4^ Data were calculated as log_2_fold change between probiotic and control treatments and expressed in fold change using a logarithmic scale to base 2.

**Table 5 animals-13-01970-t005:** Prediction of growth performance of broiler chickens at starter (weeks 1–3), finisher (weeks 4–6), and overall (weeks 1–6) periods without pathogen challenge.

Response Variable (Y) ^1^	n_Treat_	Parameter Estimates				Model Statistics		
		Intercept	SE_Intercept_	Slope	SE_Slope_	RMSE	R^2^	*p*-Value
Starter, Week 1–3								
ADFI (g)	33	48.79	2.405	0.253	0.334	8.736	0.02	0.455
ADG (g)	30	32.52	1.267	0.215	0.176	4.425	0.05	0.234
FCR	33	1.46	0.050	−0.002	0.007	0.180	0	0.741
Finisher, Week 4–6								
ADFI (g)	29	148.29	7.481	0.600	1.025	25.025	0.01	0.563
ADG (g)	26	73.80	4.703	0.661	0.644	15.012	0.04	0.315
FCR	29	1.99	0.057	−0.006	0.008	0.189	0.02	0.422
Overall, Week 1–6								
ADFI (g)	32	97.98	4.503	0.443	0.624	15.736	0.02	0.483
ADG (g)	26	53.54	2.414	0.514	0.333	7.705	0.09	0.135
FCR	32	1.77	0.030	−0.007	0.004	0.105	0.07	0.132

n_Treat_, number of treatments means; SE, standard error; RMSE, root mean square error; ADFI, average daily feed intake; ADG, average daily weight gain; FCR, feed conversion ratio. ^1^ Probiotic genera included for these response variables were *Bacillus*, *Bifidobacterium*, *Lactobacillus*, *Clostridium*, *Enterococcus*, and *Saccharomyces*.

**Table 6 animals-13-01970-t006:** Prediction of growth performance of broiler chickens at starter (week 1–3), finisher (week 4–6), and overall (week 1–6) periods with pathogen challenge.

Response Variable (Y) ^1,2,3^	n_Treat_	Parameter Estimates				Model Statistics		
		Intercept	SE_Intercept_	Slope	SE_Slope_	RMSE	R^2^	*p*-Value
Starter, Week 1–3								
ADFI (g)	22	51.82	2.483	0.132	0.347	7.456	0.01	0.709
ADG (g)	22	34.72	2.260	0.270	0.316	6.787	0.04	0.403
FCR	22	1.49	0.066	−0.007	0.009	0.198	0.03	0.464
Finisher, Week 4–6								
ADFI (g)	13	142.90	16.708	1.388	2.474	40.945	0.03	0.586
ADG (g)	13	68.16	11.278	0.758	1.670	27.638	0.02	0.659
FCR	13	2.04	0.135	−0.003	0.020	0.330	0	0.879
Overall, Week 1–6								
ADFI (g)	17	95.08	4.156	0.102	0.634	11.776	0	0.874
ADG (g)	17	55.76	4.857	0.259	0.741	13.760	0.01	0.731
FCR	17	1.75	0.091	−0.007	0.014	0.257	0.02	0.626

n_Treat_, number of treatment means; SE, standard error; RMSE, root mean square error; ADFI, average daily feed intake; ADG, average daily weight gain; FCR, feed conversion ratio. ^1.^ Probiotic genera included for these response variables were *Bacillus*, *Bifidobacterium*, *Lactobacillus*, *Enterococcus*, and *Saccharomyces.* ^2.^ Pathogens included for these response variables were *E. coli*, *C. perfringens*, *S. enteritidis*, *E. maxima*, *E. tenella*, *E. acervulina*, *E. mivati*, *E. mitis*, and *E. praecox.*
^3.^ Means of days post-infection for the starter, finisher, and overall periods were 10.8, 33.2, and 34.9 days, respectively.

**Table 7 animals-13-01970-t007:** Best-fit equations showing the response variables of gut barrier function-related gene expression and histomorphology (fold change) in relation to increasing dietary probiotics, metabolizable energy, and crude protein level in broiler chickens without pathogen challenge using backward elimination technique.

Response Variable (Y) ^1,2^	Predictor (X)	n_Treat_	Parameter Estimates				Model Statistics			
			Intercept	SE_Intercept_	Slope	SE_Slope_	RMSE	R^2^	VIF	*p*-Value
Jejunum, Week 3										
*MUC2*		10	0.95	0.149			0.336	0.40		
	Probiotic (CFU/kg)				0.053	0.023			1.00	0.050
*ZO1*		11	0.99	0.046			0.101	0.49		
	Probiotic (CFU/kg)				0.019	0.007			1.00	0.016
*OCLN*		13	0.95	0.233			0.594	0.36		
	Probiotic (CFU/kg)				0.009	0.004			1.00	0.030
*CLDN1*		11	7.37	2.871			0.209	0.62		
	Dietary ME (MJ/kg)				−0.514	0.231			1.00	0.057
	Probiotic (CFU/kg)				0.039	0.014			1.00	0.026
Villus Height		15	−0.83	0.946			0.073	0.74		
	Dietary ME (MJ/kg)				0.146	0.076			1.00	0.077
	Probiotic (CFU/kg)				0.022	0.004			1.00	0.000
Villus Height/Crypt Depth		15	−4.69	1.627			0.125	0.72		
	Dietary ME (MJ/kg)				0.456	0.130			1.00	0.004
	Probiotic (CFU/kg)				0.029	0.007			1.00	0.001
Jejunum, Week 6										
*MUC2*		10	26.72	9.897			0.688	0.70		
	Dietary ME (MJ/kg)				−1.961	0.754			1.00	0.035
	Probiotic (CFU/kg)				0.152	0.050			1.00	0.018
*ZO1*		14	0.34	4.308			0.309	0.70		
	Dietary ME (MJ/kg)				0.749	0.325			1.20	0.044
	Dietary CP (%)				−0.460	0.182			1.19	0.030
	Probiotic (CFU/kg)				0.071	0.018			1.01	0.003
*OCLN*		16	−13.01	3.829			0.440	0.62		
	Dietary CP (%)				0.702	0.193			1.00	0.003
	Probiotic (CFU/kg)				0.072	0.023			1.00	0.009
Villus Height		19	1.00	0.032			0.084	0.28		
	Probiotic (CFU/kg)				0.011	0.004			1.00	0.020
Ileum, Week 3										
*MUC2*		10	0.88	0.217			0.439	0.57		
	Probiotic (CFU/kg)				0.095	0.030			1.00	0.012
*OCLN*		13	0.85	0.150			0.338	0.47		
	Probiotic (CFU/kg)				0.064	0.020			1.00	0.009
*CLDN1*		10	0.96	0.115			0.233	0.39		
	Probiotic (CFU/kg)				0.036	0.016			1.00	0.054
Crypt Depth		11	1.00	0.058			0.130	0.29		
	Probiotic (CFU/kg)				−0.016	0.008			1.00	0.088
Villus Height/Crypt Depth		11	−4.09	2.538			0.122	0.68		
	Dietary ME (MJ/kg)				0.856	0.348			3.20	0.044
	Dietary CP (%)				−0.256	0.129			3.20	0.087
	Probiotic (CFU/kg)				0.023	0.008			1.00	0.021
Ileum, Week 6										
*ZO1*		14	57.76	14.077			0.582	0.76		
	Dietary ME (MJ/kg)				−3.390	0.755			1.71	0.001
	Dietary CP (%)				−0.695	0.299			1.72	0.043
	Probiotic (CFU/kg)				0.116	0.039			1.02	0.014
*OCLN*		16	7.27	2.943			0.231	0.45		
	Dietary ME (MJ/kg)				−0.489	0.228			1.02	0.052
	Probiotic (CFU/kg)				0.031	0.014			1.02	0.045
Villus Height		17	−0.02	0.382			0.049	0.72		
	Dietary CP (%)				0.052	0.019			1.00	0.019
	Probiotic (CFU/kg)				0.014	0.003			1.00	<0.001
Villus Height/Crypt Depth		17	2.49	0.603			0.048	0.76		
	Dietary ME (MJ/kg)				−0.114	0.046			1.01	0.028
	Probiotic (CFU/kg)				0.015	0.003			1.01	<0.001

n_Treat_, number of treatment means; SE, standard error; RMSE, root mean square error; VIF, variance inflation factor; ME, metabolizable energy; CP, crude protein; *MUC2*, mucin-2; *ZO1*, zonula occludens-1; *OCLN*, occludin; *CLDN1*, claudin-1. ^1^ Probiotic genera included for these response variables were *Bacillus*, *Bifidobacterium*, *Lactobacillus*, *Clostridium*, *Enterococcus*, *Pediococcus*, *Paenibacillus*, and *Saccharomyces*. ^2^ Data were calculated as log_2_fold change between probiotic and control treatments and expressed in fold change using a logarithmic scale to base 2.

**Table 8 animals-13-01970-t008:** Best-fit equations showing the gut barrier and immune-related gene expression (fold change) in relation to increasing levels of dietary probiotics, metabolizable energy, and crude protein, as well as days post-infection in broiler chickens with pathogen challenge using backward elimination technique.

Response Variable (Y) ^1,2,3,4^	Predictor (X)	n_Treat_	Parameter Estimates				Model Statistics			
			Intercept	SE_Intercept_	Slope	SE_Slope_	RMSE	R^2^	VIF	*p*-Value
Jejunum, Week 2										
*ZO1*		14	6.23	1.644			0.048	0.81		
	Dietary ME (MJ/kg)				−0.512	0.145			1.49	0.005
	Dietary CP (%)				0.054	0.016			1.50	0.006
	Probiotic (CFU/kg)				0.015	0.003			1.00	0.000
*CLDN3*		10	1.00	0.040			0.089	0.97		
	Probiotic (CFU/kg)				0.103	0.007			1.00	<0.001
*IL1B*		10	1.00	0.015			0.035	0.63		
	Probiotic (CFU/kg)				−0.009	0.003			1.00	0.006
*IFNG*		10	1.00	0.037			0.083	0.82		
	Probiotic (CFU/kg)				−0.037	0.006			1.00	0.000
Jejunum, Week 3										
*ZO1*		17	14.06	3.798			0.201	0.64		
	Dietary ME (MJ/kg)				−0.517	0.190			1.26	0.018
	Dietary CP (%)				−0.316	0.099			1.26	0.007
	Probiotic (CFU/kg)				0.032	0.011			1.01	0.012
*OCLN*		17	9.41	3.919			0.424	0.46		
	Dietary CP (%)				−0.404	0.187			1.00	0.048
	Probiotic (CFU/kg)				0.060	0.023			1.00	0.020
*IL1B*		17	−2.10	1.215			0.169	0.68		
	Dietary CP (%)				0.146	0.057			1.01	0.023
	Probiotic (CFU/kg)				−0.040	0.009			1.01	0.001
*IL6*		12	−4.66	1.799			0.145	0.77		
	Days post-infection				−0.022	0.008			1.12	0.021
	Dietary CP (%)				0.285	0.088			1.11	0.012
	Probiotic (CFU/kg)				−0.034	0.009			1.01	0.005
*IL10*		13	0.35	0.382			0.637	0.67		
	Days post-infection				0.066	0.033			1.01	0.072
	Probiotic (CFU/kg)				0.057	0.038			1.01	0.002
*TNFA*		10	1.01	0.066			0.150	0.46		
	Probiotic (CFU/kg)				−0.026	0.010			1.00	0.033
Jejunum, Week 4										
*ZO1*		12	11.29	2.383			0.113	0.83		
	Days post-infection				0.018	0.008			2.07	0.066
	Dietary ME (MJ/kg)				−0.828	0.194			2.05	0.003
	Probiotic (CFU/kg)				0.032	0.008			1.02	0.003
*IL1B*		10	5.03	1.414			0.164	0.87		
	Dietary CP (%)				−0.204	0.071			1.00	0.024
	Probiotic (CFU/kg)				0.076	0.012			1.00	0.000
Ileum, Week 2										
*TLR4*		10	−2.07	0.604			0.052	0.95		
	Dietary CP (%)				0.143	0.028			1.00	0.001
	Probiotic (CFU/kg)				−0.035	0.004			1.00	<0.001
Ileum, Week 3										
*ZO1*		16	11.94	4.302			0.409	0.46		
	Dietary CP (%)				−0.514	0.201			1.00	0.024
	Probiotic (CFU/kg)				0.048	0.024			1.00	0.063
*OCLN*		16	12.02	4.638			0.441	0.45		
	Dietary CP (%)				−0.517	0.217			1.00	0.033
	Probiotic (CFU/kg)				0.054	0.025			1.00	0.052
*IFNG*		12	1.01	0.043			0.097	0.71		
	Probiotic (CFU/kg)				−0.032	0.006			1.00	0.001
Ileum, Week 4										
*ZO1*		11	3.67	0.826			0.098	0.87		
	Dietary ME (MJ/kg)				−0.208	0.064			1.08	0.012
	Probiotic (CFU/kg)				0.036	0.006			1.08	0.001
*OCLN*		11	7.67	1.803			0.208	0.84		
	Dietary CP (%)				−0.335	0.089			1.06	0.006
	Probiotic (CFU/kg)				0.058	0.014			1.06	0.003
*CLDN1*		11	12.08	3.798			0.438	0.69		
	Dietary CP (%)				−0.557	0.189			1.06	0.018
	Probiotic (CFU/kg)				0.062	0.029			1.06	0.061
Ceca, Week 2										
*IL6*		18	0.96	0.093			0.270	0.31		
	Probiotic (CFU/kg)				−0.034	0.013			1.00	0.017
*IL10*		10	46.42	16.880			0.735	0.79		
	Days post-infection				0.506	0.177			1.96	0.029
	Dietary ME (MJ/kg)				−3.868	1.412			1.97	0.034
	Probiotic (CFU/kg)				0.200	0.059			1.01	0.015
Ceca, Week 4										
*ZO1*		10	−4.26	0.721			0.147	0.96		
	Days post-infection				0.727	0.095			1.00	0.0001
	Probiotic (CFU/kg)				0.014	0.001			1.00	<0.001

n_Treat_, number of treatment means; SE, standard error; RMSE, root mean square error; VIF, variance inflation factor; ME, metabolizable energy; CP, crude protein; *MUC2*, mucin-2; *ZO1*, zonula occludens-1; *OCLN*, Occludin; *CLDN1*,-3, claudin-1,-3; *IL6*, *-10*, *-1B*, interleukin-6, -10, -1beta; *TLR4*, Toll-like receptor-4; *IFNG*, interferon-gamma; *TNFA*, tumor necrosis factor-alpha. ^1.^ Probiotic genera included for these response variables were *Bacillus*, *Bifidobacterium*, *Lactobacillus*, *Paenibacillus*, *Clostridium*, *Enterococcus*, *Pediococcus*, *Streptococcus*, and *Saccharomyces.*
^2.^ Pathogens included for these response variables were *E. coli*, *C. perfringens*, *S. enteritidis*, *E. maxima*, *E. tenella*, *E. acervulina*, *E. mivati*, *E. brunetti*, *E. mitis*, *E. praecox*, *F. graminearum*, *S. pullorum*, *S. minnesota*, *L. monocytogenes*, and Aflatoxin B1. ^3.^ Means of days post-infection ranged from 3.4 to 17.3 days for various ages and gut segments. ^4.^ Data were calculated as log_2_fold change between probiotic and control treatments and expressed in fold change using a logarithmic scale to base 2.

**Table 9 animals-13-01970-t009:** Best-fit equations showing the gut histomorphology response variables (fold change) in relation to increasing levels of dietary probiotics, metabolizable energy, and crude protein, as well as days post-infection in broiler chickens with pathogen challenge using backward elimination technique.

Response Variable (Y) ^1,2,3,4^	Predictor (X)	n_Treat_	Parameter Estimates				Model Statistics			
			Intercept	SE_Intercept_	Slope	SE_Slope_	RMSE	R^2^	VIF	*p*-Value
Duodenum, Week 5										
Villus Height		15	0.78	0.073			0.035	0.75		
	Days post-infection				0.007	0.002			1.02	0.007
	Probiotic (CFU/kg)				0.012	0.003			1.02	0.001
Jejunum, Week 2										
Crypt Depth		11	0.14	0.261			0.032	0.88		
	Dietary CP (%)				0.040	0.012			1.00	0.011
	Probiotic (CFU/kg)				−0.013	0.002			1.00	0.000
Villus Height/Crypt Depth		11	0.98	0.088			0.197	0.38		
	Probiotic (CFU/kg)				0.029	0.013			1.00	0.044
Jejunum, Week 3										
Villus Height		17	−3.67	1.782			0.072	0.77		
	Days post-infection				−0.007	0.004			1.26	0.070
	Dietary ME (MJ/kg)				0.385	0.144			1.25	0.019
	Probiotic (CFU/kg)				0.019	0.004			1.02	0.000
Crypt Depth		17	−1.06	0.828			0.080	0.51		
	Dietary CP (%)				0.098	0.039			1.01	0.026
	Probiotic (CFU/kg)				−0.011	0.004			1.01	0.020
Villus Height/Crypt Depth		17	6.18	1.979			0.191	0.60		
	Dietary CP (%)				−0.247	0.094			1.01	0.020
	Probiotic (CFU/kg)				0.036	0.010			1.01	0.004
Jejunum, Week 5										
Crypt Depth		17	−2.85	1.084			0.037	0.62		
	Dietary ME (MJ/kg)				0.301	0.085			1.00	0.003
	Probiotic (CFU/kg)				−0.008	0.002			1.00	0.005
Villus Height/Crypt Depth		17	1.20	0.067			0.048	0.57		
	Days post-infection				−0.007	0.002			1.07	0.010
	Probiotic (CFU/kg)				0.012	0.003			1.07	0.002
Ileum, Week 3										
Villus Height		13	0.95	0.019			0.029	0.74		
	Days post-infection				0.006	0.001			1.00	0.002
	Probiotic (CFU/kg)				0.006	0.002			1.00	0.010
Ileum, Week 5										
Crypt Depth		15	1.28	0.072			0.034	0.75		
	Days post-infection				−0.009	0.002			1.02	0.001
	Probiotic (CFU/kg)				0.012	0.002			1.02	0.001
Villus Height/Crypt Depth		15	1.00	0.025			0.043	0.41		
	Probiotic (CFU/kg)				−0.009	0.003			1.00	0.011

n_Treat_, number of treatment means; SE, standard error; RMSE, root mean square error; VIF, variance inflation factor; ME, metabolizable energy; CP, crude protein. ^1^ Probiotic genera included for these response variables were *Bacillus*, *Lactobacillus*, *Enterococcus*, and *Saccharomyces*. ^2.^ Pathogens included for these response variables were *E. coli*, *C. perfringens*, *S. enteritidis*, *E. maxima*, *E. tenella*, *E. acervulina*, *E. mitis*, *E. praecox*, and *F. graminearum.*
^3.^ Means of days post-infection ranged from 5.2 to 32.0 days for various ages and gut segments. ^4^ Data were calculated as log_2_fold change between probiotic and control treatments and expressed in fold change using a logarithmic scale to base 2.

## Data Availability

Not applicable.

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
