# Peer review of "Dietary Probiotics Modulate Gut Barrier and Immune-Related Gene Expression and Histomorphology in Broiler Chickens under Non- and Pathogen-Challenged Conditions: A Meta-Analysis"

_animals, 2023, doi:10.3390/ani13121970_

Round 1
Reviewer 1 Report
This manuscript builds on previous studies using a meta-analysis to assess the effects of dietary probiotics on intestinal barrier and immune-related gene expression, histomorphology, and growth in broilers with or without pathogen attack. The results showed that probiotics could improve intestinal health on both structural and genetic levels, which provided a theoretical basis for more rational application of probiotics in animal husbandry in the future. In general, the subject is of interest, and this study has valuable data and results which are useful in this field. Therefore, I think this paper is well worth and suggest to accept the paper for publication with following a few minor corrections.
1. Please improve the format of this manuscript according to journal instructions.
2. Please make appropriate modifications to the abstract conclusion of the manuscript
3. The English of the manuscript needs to be properly polished
The overall English quality of this manuscript is acceptable, but some details still need to be further improved
Author Response
AUTHORS: Thank you very much for your comments. Hopefully we could address the issues sufficiently. To note, the line numbers refer to the document with track changes the visible inline (“All Markup”).
- Please improve the format of this manuscript according to journal instructions.
AUTHORS: We checked the manuscript for inconsistencies.
- Please make appropriate modifications to the abstract conclusion of the manuscript.
AUTHORS: Done as suggested (L42-43)
- The English of the manuscript needs to be properly polished.
AUTHORS: We hope that we could improve the quality of the English.
Reviewer 2 Report
The manuscript presents meta-analysis of dietary probiotics effects on gut health and histomorphology in broiler chicken. The manuscript is very well written and prepared. The subject is of interest for poultry researchers.
It would be great if there were figure or two included in the manuscript.
Author Response
AUTHORS: Thank you very much for your kind review. We have added figures to illustrate the preparation of the datasets. To note, the line numbers refer to the document with track changes the visible inline (“All Markup”).
Reviewer 3 Report
Please see minor comments in the document. This is a very interesting study; worth reading.
The meta-analyses was critically done, but specific detail (i.e., quantitative data) on the considered parameters is missing. For instance a range on a number of replicates (6 to 8 or CP to ME ratios), instead of only describing the parameters. Chick mortality and/or necropsy reports at different stages was not considered - any reasons. Also, on ingredient and dietary composition, it is not clear if any of these studies noted any bioactive compounds that could have influenced the probiotic effects, besides the CP and ME levels?
Author Response
AUTHORS: Thank you very much for your comments. To note, the line numbers refer to the document with track changes the visible inline (“All Markup”).
For instance a range on a number of replicates (6 to 8 or CP to ME ratios), instead of only describing the parameters.
AUTHORS: Table S1 shows the ranges for parameters that were extracted from the original studies. The number of replicates was provided in each Table as “nTreat = number of treatments means included”.
Chick mortality and/or necropsy reports at different stages was not considered - any reasons.
AUTHORS: Thank you for commenting on that. Our major focus of the present meta-analysis was on the effect of the probiotics on the gut (and performance). We agree, it would be worth to investigate the effect of probiotics on chick mortality.
Also, on ingredient and dietary composition, it is not clear if any of these studies noted any bioactive compounds that could have influenced the probiotic effects, besides the CP and ME levels?
AUTHORS: This is a very good point that you mention. We added the information that no other bioactive compounds were supplemented in the probiotic diets from the individual original studies to the description of the database in the Results section (L239-244). In studies in which also other bioactive components were tested, only the results for the chicken groups that received the control diet and probiotic diet were included in the dataset.